# Combination of Fetal Fraction Estimators Based on Fragment Lengths and Fragment Counts in Non-Invasive Prenatal Testing

**DOI:** 10.3390/ijms20163959

**Published:** 2019-08-14

**Authors:** Juraj Gazdarica, Rastislav Hekel, Jaroslav Budis, Marcel Kucharik, Frantisek Duris, Jan Radvanszky, Jan Turna, Tomas Szemes

**Affiliations:** 1Geneton Ltd., Bratislava 84104, Slovakia; 2Department of Molecular Biology, Faculty of Natural Sciences, Comenius University, Bratislava 84104, Slovakia; 3Slovak Centre of Scientific and Technical Information, Bratislava 81104, Slovakia; 4Comenius University Science Park, Bratislava 84104, Slovakia; 5Department of Computer Science, Faculty of Mathematics, Physics and Informatics, Comenius University, Bratislava 84248, Slovakia; 6Institute of Clinical and Translational Research, Biomedical Research Center, Slovak Academy of Sciences, Bratislava 84505, Slovakia

**Keywords:** NIPT, fetal fraction, statistical methods, DNA, maternal serum screening, fetal cells

## Abstract

The reliability of non-invasive prenatal testing is highly dependent on accurate estimation of fetal fraction. Several methods have been proposed up to date, utilizing different attributes of analyzed genomic material, for example length and genomic location of sequenced DNA fragments. These two sources of information are relatively unrelated, but so far, there have been no published attempts to combine them to get an improved predictor. We collected 2454 single euploid male fetus samples from women undergoing NIPT testing. Fetal fractions were calculated using several proposed predictors and the state-of-the-art SeqFF method. Predictions were compared with the reference Y-based method. We demonstrate that prediction based on length of sequenced DNA fragments may achieve nearly the same precision as the state-of-the-art methods based on their genomic locations. We also show that combination of several sample attributes leads to a predictor that has superior prediction accuracy over any single approach. Finally, appropriate weighting of samples in the training process may achieve higher accuracy for samples with low fetal fraction and so allow more reliability for subsequent testing for genomic aberrations. We propose several improvements in fetal fraction estimation with a special focus on the samples most prone to wrong conclusion.

## 1. Introduction

Prenatal testing for genomic defects of a fetus before birth is an integral component of current obstetric practice [1]. The primary aim of the testing is screening for the presence of abnormal number of chromosomes. Testing is focused mainly on identifying an additional copy of a chromosome, the condition called trisomy [2], represented as an excessive proportion of genomic material from the aberrant chromosome. Discovery of fetal DNA fragments in plasma extracted from maternal blood, cell-free fetal DNA (cffDNA) [3], opened up new options in the field of prenatal screening called non-invasive prenatal testing (NIPT) [1,4]. In contrast to established screening methods [5], sampling of genetic material from the mother’s circulation does not pose any direct risk for the fetus [6]. On the other hand, fetal DNA fragments constitute only a minor part of the sampled, mostly maternal, genomic material and so pose a challenge for reliable detection of present aberration.

The proportion of fetal fragments in analyzed DNA mixture is called fetal fraction (FF). A sample with a trisomic fetus typically has an aberrant proportion of genomic material from the trisomic chromosome, compared to healthy samples. A trisomic sample with a very low FF can, however, be incorrectly evaluated as healthy, since the aberrant chromosome may cause only a weak deviation from normal values that can be presumed to represent a normal measurement error [7], which is particularly problematic for the detection of sub-chromosomal aberrations. Reliable estimation of the FF is therefore a crucial step of NIPT analysis to avoid false-negative results [8,9]. This can be based on relevant anthropometric and laboratory processing attributes with significant correlation with FF, namely the gestational age and the body mass index of the mother [10]. They were, however, reported to be too weak for use as stand-alone information for reliable conclusions, and so more sophisticated technical parameters exploiting different characteristics between maternal and fetal DNA fragments are typically used [11].

Count based methods of FF determination calculate disproportion of number of reads mapped to chromosomes that differ in mother and fetus genotypes. Although they are quite reliable, they can be used only on samples with male (XX vs. XY genotype) [12] or trisomic (2 copies vs. 3 copies of an aberrant chromosome) fetuses [13,14]. In pregnancy with a healthy female fetus, the FF have to be estimated by alternative methods.

Methods based on sequence variation in DNA fragments are, on the other hand, highly reliable [15,16], although their application typically requires another laboratory assay to recover genetic map of parents, and so it is considered too expensive and time-consuming for routine diagnostics. Alternatively, FF may be estimated from the genomic location patterns that slightly differ between maternal and fetal fragments due to differences of their nucleosome positioning (SANEFALCON method) [17] and euchromatic DNA structure (SeqFF method) [18]. Due to high accuracy and no additional laboratory costs, the SeqFF method is a preferred method for samples with female fetus [19].

The length of DNA fragment is another promising attribute for differentiation of DNA fragments, since fetal fragments tend to be shorter than maternal ones (Figure 1). Although the fragment lengths improve prediction of NIPT tests by detection of false positive predictions [20,21,22], their potential for prediction of FF has been understudied. Only a simple length-based metric has been proposed in [23], where the ratio of the number of shorter (100–150 bases) and longer (163–168 bases) fragments has correlated with the FF.

We demonstrate in this study that fragment length profile may be utilized as a fairly accurate predictor of the FF. We investigate various methods and enhancements that lead to improvement of prediction accuracy and propose weighting training samples to improve accuracy in samples with low FF that are more prone to wrong testing conclusions. Although the proposed method is not as accurate as SeqFF, we show that their combination leads to more a reliable predictor than individual methods alone.

## 2. Results

### 2.1. Comparison of Length-Based Methods

We introduced a new method for estimating FF (NLRM) and also compared other statistical (LRM) and machine learning methods (NN, SVM) (see FF ESTIMATORS BASED ON FRAGMENT LENGTH in Materials and methods). The Y-based FF was used as a “ground truth” for each comparison. We used Pearson correlation and mean squared errors (MSE) of the predicted FF and the Y-based FF to compare the surveyed methods.

Correlations were higher when compared to ones obtained from a published method using the ratio of shorter and longer fragments (median correlation 0.568 on our data) [23]. After picking the best ratio of consecutive intervals of fragment lengths (FRAC ratio), we were able to significantly improve this correlation to 0.728. To get even better results, we applied nonlinear regression model (NLRM) to estimate parameters (or weights) for individual fragment lengths resulting from the FRAC ratio. Here we observed median correlation 0.812 for test set.

Although NN initially seemed to be a promising method for estimating FF using fragment length distribution, we observed only sub-par results (median correlation 0.778). LRM and SVM models achieved best correlation with median value 0.83 and 0.831, respectively. All results can be seen on Figure 2. 

### 2.2. Combination of Methods Improves Prediction Accuracy

We have chosen the SVM method (median correlation 0.831) as the representative method based on fragment-length profiles, according to the best prediction accuracy in the previous comparison. SeqFF performed slightly better on our dataset than the SVM method (0.831 vs. 0.877, see Figure 3). Since these two predictors use different attributes of fragments for prediction (lengths and positions), we combined these two approaches using a linear regression model.

Furthermore, we examined the effect of additional predictive attributes from relevant anthropometric and laboratory processing attributes, namely the gestational age, the body mass index of the mother and the DNA library concentration to improve the models. In all examined conditions, the accuracy of the predictions was improved, although the improvement was only mild in some cases (Figure 3, with and without SA). Finally, using a linear regression model, we combined two approaches, SVM and SeqFF, whereby the best trained model almost reached 0.95 correlation when testing. Comparison of tested data from a randomly selected combined model with a Y-based method is presented in the supplement (Figure 4) showing no systematic difference between the two methods. Moreover, we compared FF prediction between male and female samples using the combined method. Similar distribution of male and female samples was observed (Appendix A).

Finally, we provide trained attribute (i.e., feature) weights of four examined linear models with different combinations of attributes in a separate table (Table 1).

### 2.3. Weighting of Samples with Low Fraction

From the perspective of NIPT, it is crucial to decrease the probability of false-negative results to a minimum. Thus, only results with predicted FF above a certain threshold are acceptable. In this sense, we examined adjustment of the model that would improve prediction accuracy on critical, low fraction samples (FF <10%) at the expense of prediction of high fraction samples (FF >10%). We achieved the improvement with inclusion of multiple copies of low fraction samples in training, increasing the significance of such samples on resulting parameters.

According to the expectation, we observed decline of prediction accuracy of samples with high fraction. On the other hand, the prediction accuracy of low fraction samples was gradually improved with increased significance of considered samples (Figure 5). This suggests that weighting samples may further improve the decision process with more precise prediction of critical samples.

## 3. Discussion

Accurate estimation of fetal fraction in analyzed mixtures of fetal and maternal DNA fragments is a critical part of non-invasive prenatal testing. The main challenge is to identify samples with too low a number of fetal fragments for reliable prediction of aneuploidy, since over-estimation of prediction may lead to false-negative results. Conversely, accuracy is not so important in typical settings for samples with a sufficiently high fetal fraction. Prediction models should be therefore trained and evaluated with a special focus on the importance of individual samples.

Several methods have been proposed for FF prediction, each with its own benefits and limitations. Early established methods were based on disproportion of number of fragments from sex chromosomes. They are simple to understand and implement and require a relatively small number of samples for training. Their calculation is therefore typically the first step in estimation, whether as the gold standard for other trained methods or initial determination of fetus sex. Testing may also utilize two separated prediction methods for male and female fetuses, possibly with different thresholds based on their precision. With the rising number of tested samples, the more complex models can be trained without the risk of overfitting. Since the number of samples should be much higher than the number of trained model parameters, laboratories should reach at least hundreds of samples for fragment length based models and tens of thousands for the position based models. This is a great advantage for simpler models, since the acquisition of such a number of samples may be limiting for small laboratories. This represents a problem for already established tests as well, since beneficial changes in laboratory processing may be halted due to a high cost of re-training.

Available methods are also highly diverse in the attributes they use for prediction, from the simplest models that utilize only patient attributes, to highly complex attributes aggregated from sequenced fragments. Combination of several independent, even weak predictors, can perform better than any of them alone, as was shown in many other domains [24]. Fetal fraction prediction benefited from such coupling as well, as presented in this study. Furthermore, the more accurate predictions including more methods may indicate or even correct the potential abnormal diversions of any single method. Present or future methods may be incrementally supplemented and so gradually improve the overall predictions. The effect of individual methods should be however weighted to prefer methods with higher accuracy. Assigned weights may also indicate that a gain from a less accurate predictor becomes negligible and so may be excluded without significant impact.

We propose several approaches that markedly improve accuracy of prediction of fetal fraction, making results of non-invasive prenatal testing more reliable. Firstly, the fragment length profiles may be utilized as a reliable predictor of FF and achieve similar precision to the favored methods based on positions of DNA fragments. Secondly, the combination of results from multiple predictors achieve far better predictions, at least when they are based on different attributes of input DNA fragments. The final combined method is free of systematic bias when compared to the Y-based method. We conclude that low quality predictors, like regressors based on relevant anthropometric and laboratory processing attributes, may contribute to the overall accuracy. Other methods utilizing their own set of distinctive attributes, both present and future, can be similarly included, and so improve prediction compared to their stand-alone usage. Based on these findings, we conclude that additional, possibly independent, information can significantly raise the prediction accuracy of FF prediction, and thus, should be used when possible.

Since the SeqFF publication does not provide a training option, we were forced to use the published parameters on our dataset. As a consequence, we were unable to reach correlation reported by the SeqFF study; however, our combined method reached similar scores. We presume that the combined method can significantly surpass the performance of the standalone SeqFF method if its parameters are properly trained on a similar dataset.

Finally, appropriate weighting of samples in the training process may achieve higher accuracy for samples with low FF, and so allow a more decision regarding which samples have enough fetal fragments for subsequent testing for genomic aberrations.

## 4. Materials and Methods

### 4.1. Sample Acquisition

We collected 2454 informative samples from women undergoing NIPT testing, with single euploid male fetus, for training and testing of FF models. These samples were concluded as informative based on their fetal fraction and manual examination. The hard coded FF threshold for an informative sample was 4% but in some cases a sample with lower FF was reclassified as informative by manual examination. Our work was part of two clinical studies approved by the Ethical Committee of the Bratislava Self-Governing Region (Sabinovska ul.16, 820 05 Bratislava): the first one called “NIPT study” (study ID 35900_2015 approved on 30 April of 2015 under the decision ID 03899_2015) and the second one called “SNiPT” (study ID 37136/2018 approved on 11 June of 2018 under the decision ID 07507/2018/HF). All patients that were included in the study signed written informed consents consistent with the Helsinki declaration which were approved by the above-mentioned ethics committee.

### 4.2. Sample Preparation and Sequencing

Peripheral blood from pregnant women was collected into EDTA tubes and kept at 4 °C until plasma separation. Blood plasma was separated within 36 h after collection and stored at −20 °C at a DNA isolation unit. DNA was isolated using a QIAgen DNA Blood Mini Kit (Hilden, Germany). Standard fragment libraries were prepared from isolated DNA using a modified protocol of the Illumina TruSeq Nano Kit (San Diego, CA, USA) as described previously [20]. Briefly, to decrease the laboratory costs we used reduced volumes of reagents, which was compensated for by completing 9 cycles of PCR instead of 8, as per protocol. Physical size selection of cfDNA fragments was performed using specific volumes of magnetic beads in order to enrich FF. Illumina NextSeq 500/550 High Output Kit v2 (San Diego, CA, USA) (75 cycles) was used for massively parallel sequencing of prepared libraries using pair-end sequencing with read length of 35 bp.

### 4.3. Mapping and GC Correction

The first stage of data processing was carried out as previously described [20]. NextSeq-produced fastq files (two per sample) were directly mapped using the Bowtie 2 [25] algorithm with --very-sensitive option to the human reference genome hg19 (GRCh37). Reads with mapping quality of 40 or higher were retained for further data processing. Length of a DNA fragment was determined as the difference of the leftmost and the rightmost mapped base of the corresponding read pair.

Next, we weighted mapped reads to eliminate the GC bias according to [26] (with the exclusion of intra-run normalization) to retrieve better estimates of underlying chromosomal distributions.

The corrected number of fragments per chromosome is determined by summing the corrected read counts over all bins of the specified chromosome. The exception is chromosome Y, which is presented only pregnancies with male fetuses. Even in such cases, low proportion of mappable regions does not contain enough reads for reliable correction. As a result, a vector of corrected number of fragments per autosomes, called autosomal counts, corrected number of fragments per chromosome X (chrX), and uncorrected number of fragments per chromosome Y (chrY) were passed to the calculation of the reference, Y-based FF (see below).

### 4.4. Data Preparation and Evaluation of Models

Fetal fractions were calculated using several proposed predictors and state-of-the-art method SeqFF. Predictions were compared with the reference Y-based method. The robustness of this comparison stems from using a multitude of training and testing datasets, thereby suppressing bias from a single random sampling. Each model presented in this paper was evaluated on 100 testing sets by Pearson correlation coefficient between Y-based values and predicted values of FF. For each such experiment, the whole dataset (2454 male samples) was divided into 80% for training and 20% for testing.

The proportion of fragments for each target length (50–220 bases) were calculated for each sample separately. Moreover, in the case of the support vector machine and neural network estimators, the resulting 171 feature vectors of training sets were normalized to have zero mean and unit variance.

#### Y-Based Estimator

The Y-based FF was calculated according to the equation
(1)$$FY=\frac{\%chrY\;-\;female\%chrY}{male\%chrY\;-\;female\%chrY}$$
from [27], where male %chr Y and female %chr Y is the mean fraction of chromosome Y sequence reads of plasma samples obtained from 14 adult male individuals and pregnant women bearing euploid female fetuses, respectively. %chr Y is the fraction of chromosome Y sequence reads of the sample whose FF we want to calculate.

### 4.5. FF Estimators Based on Fragment Length

#### 4.5.1. Linear Regression Model (LRM)

Fetal fractions were estimated using a standard linear regression model provided by R-software given by the equation
(2)F=Lθ+ε,
where *F* is vector of fetal fractions, *L* is matrix of fragment lengths from 50 to 220 bp, *θ* is vector of parameters to be estimated, and *ε* is vector of errors. The training and testing sets were selected from the collected data. The parameters *θ* resulting from training were used to estimate FF for testing data.

#### 4.5.2. Non-Linear Regression Model (NLRM)

We introduce a new approach for FF prediction—nonlinear function used in NLRM. Motivation behind this method originated from a published method using the ratio of shorter (100–150 bases) and longer (163–168 bases) fragments [23]. However, when we used a ratio of the same fragment lengths as reported in the publication, we obtained sub-par results for our data. Therefore, we think that the choice of fragment lengths is data specific and, in the first step, we were exhaustively searching for the best fragment lengths to use for our data. 

Let I = $L_{50},L_{51},\ldots ,L_{220}$ be set of proportions of fragments of length 50 bp to 220 bp. Let $I_{1},I_{2}$ be subsets of consecutive elements of I. Ratio of fragments is defined as
(3)$$FRAC=\frac{\sum_{l\in l_{1}}l}{\sum_{l\in I_{2}}l}$$

To effectively compute the best performing ratio, reduction of the number of parameters is needed when using optimization methods. To this end, we chose only consecutive intervals of fragment lengths $I_{1},I_{2}$ for evaluation. The ratio with the best correlation with Y-based FF (FY) was then chosen for second step. We found out that cor(FRAC,FY) was maximal for the subsets $I_{1}=L_{131},L_{132},\ldots ,L_{134}$ and $I2=L_{97},L_{98},\ldots ,L_{153}$.

In the second step, we applied a non-linear regression method for estimating model parameters (weights) corresponding to fragment lengths. In this case, NLRM is defined by the equation
(4)F=η(θ)+ε,
where $F=\left(F_{1},\ldots ,F_{N}\right)$ is the vector of fetal fractions, η is a nonlinear function of selected fragment lengths, $\theta ={\left(M_{131},\ldots ,M_{134},N_{97},\ldots ,N_{153}\right)}^{T}$ is vector of parameters to be estimated, and ε is vector of errors. Consider the non-linear regression model where for each sample (indexed by i=1,…,N), the FF estimation is given by
(5)$$F_{i}=\frac{\sum_{r=131}^{134}M_{r}L_{r,i}}{\sum_{p=97}^{153}N_{p}L_{p,i}}+\varepsilon_{i},i=1,2,\ldots ,N.$$

Note that in the above, the nonlinear function η is given by the ratio of fragment lengths weighted by parameters $\theta ={\left(M_{131},\ldots ,M_{134},N_{97},\ldots ,N_{153}\right)}^{T}$.

Similarly to LRM, the training and testing sets were selected from the collected data. The *nlm* function from *R* software (https://www.r-project.org/) was used to estimate the parameters for NLRM. The parameters were then used to estimate FF for testing data. Other optimisation methods from *R* software like *nls* function and *optim* function using Broyden–Fletcher–Goldfarb–Shanno (BFGS) algorithm [28] were also applied to estimate parameters for NLRM, but resulting variance of the FF was much bigger compared to using *nlm.*

#### 4.5.3. Neural Network (NN)

A multilayer perceptron (MLP) regression model was trained in several steps for which the datasets of normalized fragment length proportions were preprocessed. We selected *DNNRegressor* estimator from open source machine learning framework *TensorFlow* [8] as implementation of MLP regression. We tried two- and three-layer network architectures and several activation functions in the *DNNRegressor* method. As a result, we found the following configuration to have the best performance for the given input size: 55 units in input layer and 25 in two hidden layers each, *Adagrad* optimizer, hyperbolic tangent as activation function, and mean reduction of loss.

#### 4.5.4. Support Vector Machine (SVM)

We selected a SVM estimator from open source machine learning library *sklearn* [29] as the implementation of SVM regression. Again, datasets of normalized fragment length proportions were used to predict fetal fraction. Parameters of the SVM were optimized through grid search using the method of sklearn (*GridSearchCV*) in conjunction with rigorous testing and empirical evidence. The following parameters were found to give the best results: linear kernel, epsilon 0.01, tolerance for stopping criterion (tol) 0.001, and penalty parameter of the error term (C) 1.0. Non-linear kernels were also tested but all had poorer performance than linear kernel.

### 4.6. FF Estimators Based on Read Counts

#### SeqFF

The number of fetal DNA fragments varies in different regions of the genome and significantly correlates with GC content and presence of coding regions. The position-based prediction method SeqFF [11] takes a vector of Loess-corrected fragment counts as input, partitioned into bins 50,000 bases long. The fetal fraction is then determined using standard multivariate regression models trained on a large number of samples (25,312 in original study) which exceeds this study close to tenfold. In addition, published software for SeqFF calculation does not provide a training option. For these reasons we used a pre-trained model and testing script from the original study to evaluate prediction accuracy on our dataset.

### 4.7. Correlation with Sample Attributes

Additional sample attributes from requisition were compared with the reference FF. Slight significant correlation was observed in three attributes, namely the gestational age, the body mass index of the mother, and the DNA library concentration, with Pearson correlation at the level 0.1, −0.33, and −0.22, respectively.

### 4.8. Combinaton of FF Estimators

A combined estimator for FF was built from SVM estimator and SeqFF estimator. Furthermore, we examined impact of additional predictive attributes from requisitions and laboratory processing of samples, namely the gestational age, the body mass index of the mother, and the DNA library concentration. Then, 5 variables, specifically FF estimates resulting from the SeqFF and SVM methods, the gestational age, the body mass index of the mother, and the DNA library concentration were inserted into linear regression with Y-based FF as a response variable. Finally, the parameters obtained from this linear regression were used to estimate FF for testing data.

### 4.9. Weighted Samples

The crucial step in the standard NIPT analysis is to identify samples with low fetal fractions that may not have enough fetal fragments for reliable identification of aneuploidy. Such samples are concluded as uninformative and may be subject to repeated sampling, typically few weeks later. Overestimation of FF may spuriously inflate confidence in a ploidy call for a sample where the assay is not expected to be sensitive. On the other hand, for high-FF samples the accuracy of FF is less important because the aneuploidy-calling algorithm is sufficiently sensitive anyway. 

With this fact in mind, we tried to improve predictions in the lower range of Y-based FF targets. All samples with fetal fraction below 10% (according to the Y-based estimator) within the dataset were duplicated, triplicated, and quadruplicated, giving three distinct datasets. To evaluate this approach, we trained prediction models on these datasets and compared them with models trained on the original dataset. Prediction ability of FF in lower and higher range was evaluated separately.

## Figures and Tables

**Figure 1 ijms-20-03959-f001:**
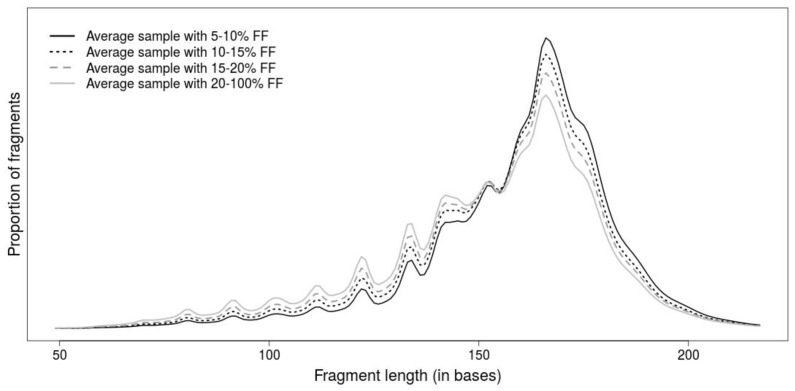
Average fragment-length profiles determined from samples with selected ranges of fetal fractions calculated across lengths. Samples with higher fetal fraction (FF) also have shorter fragments indicating that maternal fragments are longer than fetal ones.

**Figure 2 ijms-20-03959-f002:**
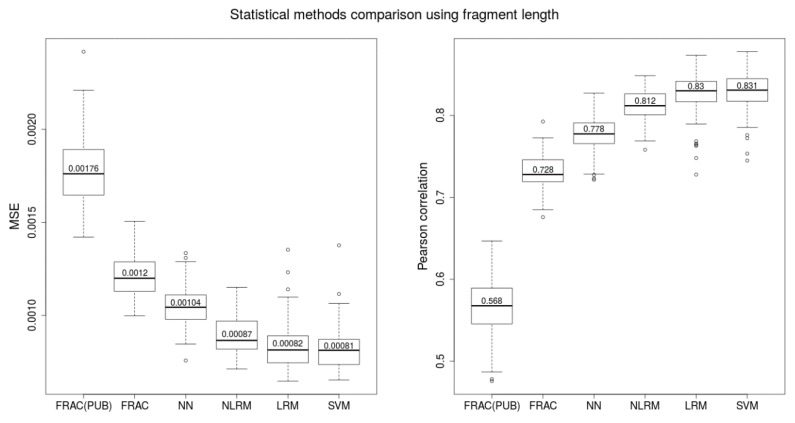
Boxplots of MSE and Pearson correlation of Y-based FF and estimated FF obtained from several different methods calculated for 100 testing sets. Training was performed on 100 complementary training sets. FRAC(PUB) represents the ratio of fragment length intervals designed by the study (Yu et al., 2014), FRAC means our best ratio of fragment length intervals derived by the best Pearson correlation with Y-based FF, NN—neural network, NLRM—non-linear regression model based on FRAC parameters, LRM—linear model using all fragment length parameters from 50–220 bp, SVM—support vector machine using all fragment length parameters from 50–220 bp.

**Figure 3 ijms-20-03959-f003:**
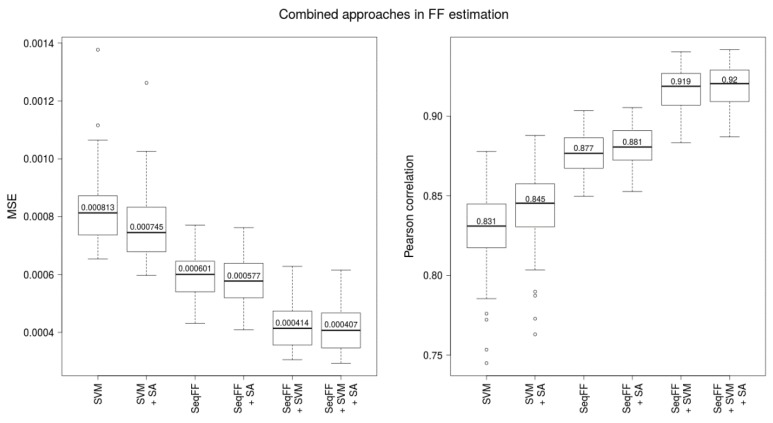
Boxplots of MSE and Pearson correlation of Y-based FF and estimated FF using SVM method calculated for 100 testing sets. Training was performed on 100 complementary training sets. Combined approach is denoted SeqFF + SVM. Improvement for every method was achieved by adding sample attributes (DNA library concentration, BMI, gestational age)—represented by “+ SA”.

**Figure 4 ijms-20-03959-f004:**
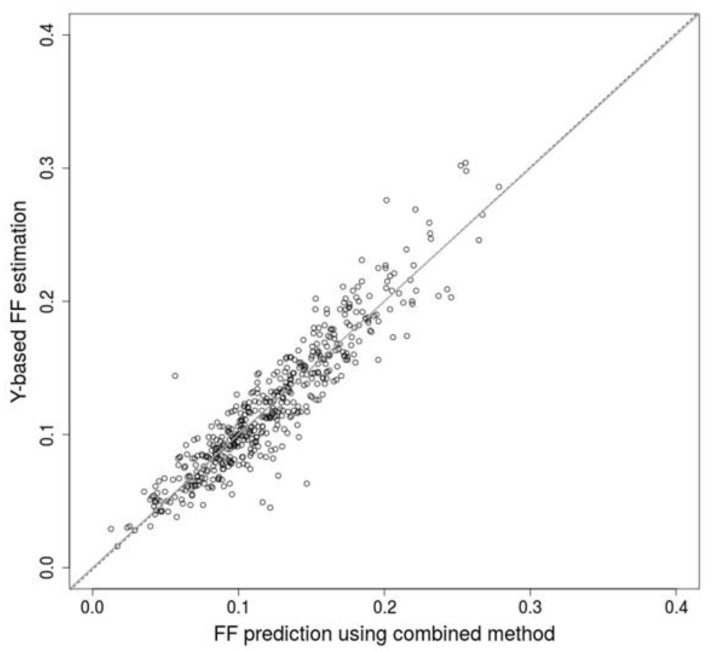
Linear regression of Y-based method with the combined method. Black circles denote individual testing samples. Dashed line represents overall trend of the prediction and the grey line is the 45° line.

**Figure 5 ijms-20-03959-f005:**
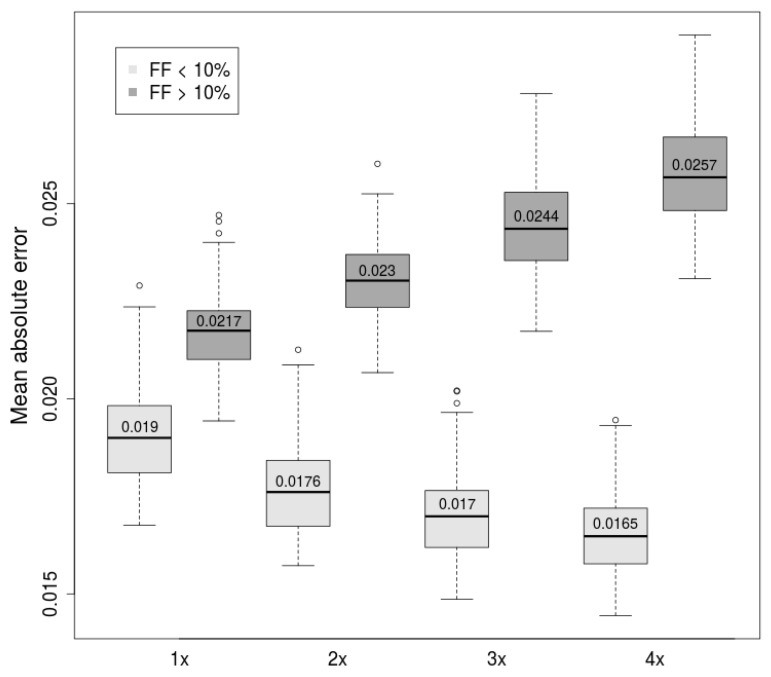
Mean absolute error Y-based FF and estimated FF using SVM method (lower is better). Only samples with FF < 10% used in training were sampled (weighted) with weights 2×, 3×, and 4×. Samples with FF > 10% were selected once. Scatterplots represented by linear regression with weighted samples are presented in the supplement (Appendix A).

**Table 1 ijms-20-03959-t001:** The table shows corresponding feature weights of four trained linear models. Each row represents a single feature of a linear model and each column represents a specific model. Each model has a different combination of features. If a feature is not part of the model, the value is empty (labeled by dash). Since one hundred models were trained for each combination, only the model with median correlation is displayed. SVM: support vector machine estimator prediction based on fragment length profile. SeqFF: prediction of the SeqFF model. BMI: body mass index of the mother. LC: DNA library concentration. GA: gestational age. SA: sample attributes (BMI + LC + GA).

Method	SeqFF + SA	SVM + SA	SVM + SeqFF	Seqff + SVM + SA
SVM	–	0.0418	0.0269	0.0243
SeqFF	1.1325	–	0.0237	0.0255
BMI	−0.0006	−0.0058	–	−0.0019
LC	−0.0020	−0.0005	–	−0.0015
GA	~0.0000	0.0037	–	0.0016
Intercept	0.0204	0.1222	0.1223	0.1227

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
