# Peer review of "Combination of Fetal Fraction Estimators Based on Fragment Lengths and Fragment Counts in Non-Invasive Prenatal Testing"

_ijms, 2019, doi:10.3390/ijms20163959_

Round 1

Reviewer 1 Report

The Manuscript “Combination of fetal fraction estimators based on fragment lengths and fragment counts in non-invasive prenatal testing” (ijms-543178) is an interesting study about the estimation of fetal fraction in non-invasive prenatal testing. In this study, the Authors investigated different methods to evaluate the fetal fraction in cfDNA non-invasive prenatal testing and combined them to get an improved predictor. The study was approved by the Ethical Committee of the Bratislava Self-Governing Region (Sabinovska) and all included patients signed informed consent.

This paper is well written, has important clinical message, and should be of great interest to the readers of International Journal of Molecular Sciences. I appreciated the methodology used by the Authors in their study. However, the Manuscript can be further expanded and improved. Therefore, according to my opinion, only a few small improvements are needed to make the article suitable for publication.

According to my opinion, only a few small improvements are needed, as suggested below:

-       I recommend a linguistic revision of the manuscript by a native English speaker in order to improve its readability.

1.       Introduction. I would suggest improving lines 38-41. It is of paramount importance highlighting that NIPT based on cfDNA is not diagnostic but is a screening test reporting a risk. Therefore, although the high sensitivity and specificity, they are not diagnostic as amniocentesis and chorionic villus sampling.

2.       The authors have not adequately highlighted the strengths and limitations of their study. I suggest better specifying these points.

3.       Discussion, I would suggest improving this section reporting which algorithm the Authors propose for male and female fetus as the best approach base on the cost-effectiveness.

4.       I appreciated the novelty of the topic and the methodological rigor of this study. However, I think it would be interesting to mention, even briefly, the clinical utility of non-invasive prenatal testing in pregnancies with ultrasound anomalies. Some useful references for this purpose are the following: PMID: 27515011; PMID: 20501973; PMID: 17353878.  

Author Response

1. Yes, we updated the manuscript accordingly.

2. In the discussion we mention three strengths of our study:

    the fragment length profiles may be utilised as a reliable predictor of FF and achieve similar precision as the favored methods based on positions of DNA fragments

    the combination of results from multiple predictors achieve far better predictions, at least when they are based on different attributes of input DNA fragments

    appropriate weighting of samples in training process may achieve higher accuracy for samples with low FF, and so allow to more reliably decide which samples have enough fetal fragments for subsequent testing for genomic aberrations

In the revised manuscript we add limitations concerning SeqFF parameters and the size of used dataset.

3. Running these algorithms is virtually free, hence they can not be compared by cost. Moreover these methods use input data obtained from the same sequencing process.

4. We looked at the proposed articles and we considered the first one to be close related but we were unable to think of relevant citation for our manuscript. The article mentiones "NIPT should not be recommended for genetic evaluation of the etiology of ultrasound anomalies" and thus can not be used as valid example of NIPT utilisation.

Reviewer 2 Report

The authors explore the possibility of combinig the seqFF estimator of fetal fraction (ff) and an estimator utilizing fragment lenght.

Comments, questions, suggestions for improvement:

1. Since there are 2454 samples, some 100 of them should have fetal fraction below the conventional LoD of 4%.
Judging by Supp figure 1, there are at most 10 such samples in the studied cohort. Why is it so?

2. How many adult male individuals were sequenced and used in computing FY, the chromosome Y-based estimate of ff?

3. How does the combined estimator of ff performs on female-carrying pregnancies, relative to seqFF? Please, provide findings.

4. As it was pointed out in [1], correlation between two methods for fetal fraction determination may be high, yet, there may be substantive systematic differences between the methods.
5. The presence or absence of systematic differences between the combined method and the Y-based ffY method needs to be discussed.

6. On p. 9, lines 306-308, it is written: "The fetal fraction is then determined using standard multivariate regression models. Due to a large number of required model parameters, a huge number of samples are required for training (25,312 training samples used in the original study) that highly exceeds scope of this study." This is misleading. seqFF employs two machine learning algorithms (the elastic network and the reduced-rank regression) for the bin selection and prediction of ff. These algorithm does not require 'huge number of samples for training'.

p. 3, l. 92, ')' is missing.
p. 9, l. 300, '... had lower performance'; consider 'had poorer performance'.

[1] Grendar M, Loderer D, Lasabova Z, Danko J, A comment on “Comparing methods for fetal fractiondetermination and quality control of NIPT samples”, Prenatal Diagnosis. 2017;37:1265.

Author Response

1. We collected 2,454 informative samples from women undergoing NIPT testing, with single euploid male fetus, for training and testing of FF models. These samples were concluded as informative based on their fetal fraction and manual examination. Hard coded FF threshold for informative sample was 4% but in some cases sample with lower FF was reclassified as informative by manual examination.

2. ChrY fraction estimate was obtained from 14 adult male individuals.

3. We compared FF prediction between male and female samples using the combined method. Similar distribution of male and female samples was observed (revised supplement figure 4-5).

4. Comparison of tested data from randomly selected combined model with Y-based method is presented in the revised supplement (figure 4) showing no systematic difference between the two methods.

5. Comparison of tested data from randomly selected combined model with Y-based method is presented in the revised supplement (figure 4) showing no systematic difference between the two methods.

6. Since the SeqFF publication does not provide training option, we were forced to use the published parameters on our dataset. As a consequence, we were unable to reach correlation reported by the SeqFF study, however our combined method reached similar scores. We presume that the combined method can significantly surpass the performance of standalone SeqFF method if its parameters are properly trained on similar dataset.

Round 2

Reviewer 2 Report

In the original version of the paper the authors state "Due to a large number of required model parameters, a huge number of samples are required for training (25,312 training samples used in the original study) that highly exceeds scope of this study". In my review, I have pointed out, that Kim et al. have used machine learning methods to train SeqFF, which do NOT require huge number of samples.  The authors could have trained their own instance of SeqFF, using their own training set of samples. --However, my objective was not to force the authors to train their own instance of SeqFF, rather, to delete their false/misleading claim; which the authors did, in the revised version.